# Electrophoretic Deposition of ZnO-Containing Bioactive Glass Coatings on AISI 316L Stainless Steel for Biomedical Applications

Farnaz Heidari Laybidi [1], Abbas Bahrami [1,*], Mohammad Saeid Abbasi [1], Mojtaba Rajabinezhad [1], Bahar Heidari Beni [1], Mohammad Reza Karampoor [1] and S. H. Mousavi Anijdan [2,*]

1    Department of Materials Engineering, Isfahan University of Technology, Isfahan 84156-83111, Iran; farnazheydari07@gmail.com (F.H.L.); msaeid.abbasi@gmail.com (M.S.A.); mojtabaa.rajabi22@gmail.com (M.R.); baharheidari295@yahoo.com (B.H.B.); mr_karampoor@yahoo.com (M.R.K.)
2    Department of Mechanical and Industrial Engineering, University of Toronto, Toronto, ON M5S 3G8, Canada
*    Correspondence: a.n.bahrami@iut.ac.ir (A.B.); hashem.mousavi@mail.mcgill.ca (S.H.M.A.); Tel.: +98-313-391-5713 (A.B.); Fax: +98-313-391-2752 (A.B.)

**Abstract:** The main objective of this investigation was to study the implications of incorporating zinc oxide nanoparticles into the matrix of a bioactive glass for the bioactivity and structural properties of the deposited coating. ZnO-containing bioactive glass was coated on an AISI 316L stainless steel substrate using the electrophoretic deposition technique. AISI 316L stainless steel is a biomedical grade steel, which is widely used in different biomedical applications. For the electrophoretic deposition, voltages and times were chosen in the range of 15–40 V and 15–120 min, respectively. The microstructure, phase composition, and surface roughness of coated samples were analyzed in this investigation. Moreover, the corrosion behavior and the MTT (mitochondrial activity) of samples were studied. Results showed a uniform distribution of elements such as silicon and calcium, characteristic of bioactive glass 58S5, in the coating as well as the uniform distribution of Zn inside the ZnO-containing samples. The findings showed that the deposited ZnO-containing bioactive glass is a hydrophilic surface with a relatively rough surface texture. The results of the MTT and antibacterial effects showed that the deposited layers have promising cell viability.

**Keywords:** bioactive glass; electrophoretic deposition; zinc oxide





## 1. Introduction

The failure of dental and orthopedic implants can be ascribed to various factors, including dynamic loadings, bacterial infection, and the lack of integration at the implant/tissue interface [1–3]. Currently employed preventive measures to minimize implant-associated failures such as using medications and sterilization of surfaces are certainly not enough. The surface of an implant is a critical interface for the tissue/implant attachments and is also a place for the formation of germs and biofilms [4]. Bone regeneration in temporary and permanent orthopedic implants largely depends on the biocompatibility of the surfaces of implants.

Stainless steels, widely used in biomedical applications, have inherently very low biocompatibility and osseointegration [5]. Moreover, stainless steel grades usually contain alloying elements and impurities that could be harmful to the surrounding tissues. Therefore, developing biocompatible coatings is extremely important for surface modification and corrosion protection of orthopedic implants. When it comes to the biomedical applications of stainless-steel grades, surface modification with bioactive layers is often inevitable. Lack of integration and implant loosening dramatically affect implant survival inside the body, resulting in reversion surgery [6–8], which is certainly a painful operation and is also

very expensive. When it comes to the integration, the interface and interfacial reactions are certainly key issues [9–11].

As has been postulated, applying a bioactive glass layer on the surface of an implant can have remarkably positive implications for the osseointegration at the interface of the implant and the bone tissue [12,13]. Bioactive glasses are known to have superior biocompatibility as compared with other biomedical materials, resulting in a perfect tissue/implant integration. Having a bioactive glass layer at the tissue/implant interface creates an environment with the ability to gradually substitute damaged bone tissues, which ultimately results in tissue regeneration with relatively slow kinetics. An added value of such a bioactive glass layer is the possibility of tuning its composition to include extra functionalities. Ion-replaced bioactive glass coatings can be designed for different purposes without any major associated toxicity and infections [14,15]. Overall, bioactive glasses can serve as a coating layer to improve the integration of metal implants into the tissue by stimulating apatite formation at the implant/tissue interface. In addition, they can minimize the risk of corrosion at the surface implants [12].

Among different nanoparticles to be added to the bioactive glass matrix, zinc oxide appears to be an excellent candidate. ZnO is known to be a biocompatible particle with great stability and hardly any toxicity. Moreover, ZnO is essentially antibacterial and very affordable, making it a promising additive for biomedical applications, especially when it comes to bacterial infections [16–18]. Surface functionalization is an important issue, as it adds extra functionalities to the surface of bioactive materials [19,20]. The AISI 316L stainless steel implants have been widely used in different orthopedic applications, on the grounds that AISI 316L alloy exhibits remarkable mechanical properties, stability, and corrosion resistance [21]. Yet, as mentioned earlier, the biocompatibility of AISI 316L grade is rather poor, raising many concerns about the implant/tissue integration. The poor surface characteristics of AISI 316L steel can cause inflammatory and immune responses from the body. Therefore, surface modification and bio-functionalization of AISI 316L steel grade have become critical issues. A cost-effective and easy-to-implement approach to enhance the bioactivity of AISI 316L steel substrates is using a modified bioactive glass as a coating layer [21,22].

Different methodologies can be employed to apply a bioactive glass-based coating layer on the surface of an implant, including enameling, sol-gel, laser cladding, thermal spray and electrophoretic deposition (EPD), among which EPD is known to be a versatile and easy-to-apply technique in biomedical applications [23]. EPD is a proper choice when depositing a uniform and relatively compact layer with a good adhesion is a concern. Additionally, EPD coatings are consistent and comparatively cheap and are applicable to complex geometries. The tuneability and control over EPD coatings are also excellent, as the chemistry of the depositing material can be very well controlled with this method [24–26]. The EPD method enables the deposition of layers comprising bioactive constituents such as cells, proteins, drugs, and bioactive polymers.

Further developments in the implant industry will certainly be focused on the surface functionalization and biomedical coating of implants, aiming to enhance their biocompatibility, bioactivity, antibacterial effects, surface wettability, and adhesion strength. This research investigates the idea of electrophoretic deposition of ZnO-containing 45S5 bioactive glass, as a cost-effective, easy-to-conduct and facile surface functionalization technique for orthopedic implants. This paper is a step forward in developing our understanding of the correlation between the kinetics and parameters of EPD and the biomedical properties of deposited ZnO-containing 45S5 bioactive glass.

## 2. Materials and Methods

### 2.1. Materials

Zinc oxide nanoparticles (n-ZnO, Sigma-Aldrich, Cologne, Germany), Nitric acid (HNO$_3$, Sigma-Aldrich, Germany), Tetraethyl orthosilicate (TEOS, Sigma-Aldrich, Germany), Triethyl phosphate (TEP, Sigma-Aldrich, Germany), Ammonia (NH$_3$, Germany),

Calcium nitrate ($Ca(NO_3)_2$, Sigma-Aldrich, Germany), deionized water, and ethanol were used to prepare composite coating samples. The ZnO content was chosen from 0 to 20 wt.%. Samples were named: BG (100 wt.% BG), BG-5%ZnO (5 wt.% ZnO and 95 wt.% BG), BG-10%ZnO (10 wt.% ZnO and 90 wt.% BG), BG-15%ZnO (15 wt.% ZnO and 85 wt.% BG), and BG-20%ZnO (20 wt.% ZnO and 80 wt.% BG). To achieve a proper dispersion of components, the suspensions were magnetically stirred for 10 min followed by 60 min of sonication.

### 2.2. Synthesis

In order to synthesize bioactive glass/ZnO composite samples, the Stöber method was used, according which two different solutions were prepared and used. The first solution comprised 2.25 g Tetraethyl orthosilicate, 0.23 g Triethyl phosphate and 25 mL ethanol. Different percentages of ZnO were added to this solution. For the second solution, 6 mL ammonia ($NH_3$), 12.38 mL water, and 8.12 mL ethanol were mixed. Finally, both solutions were mixed and stirred for 10 min. In the final step, calcium nitrate was added and the solution was stirred for 2 h at room temperature.

### 2.3. Electrophoretic Deposition (EPD)

In order to make a suspension for electrophoretic deposition, 100 mL deionized water with 0.3 g bioactive glass was prepared and stirred until a homogeneous solution was attained. Then, 180 mL acetic acid was added to the suspension. The pH of the solution was controlled at 6 with a dropwise NaOH addition. The same procedure was applied to ZnO-containing bioactive glass composites. Stainless steel AISI 316L (chemical composition: Fe-18%Cr-12%Ni-1.5%Mn-2%Mo-0.5%Si-0.05%C) plates (4 $cm^2$ area and 2 mm thickness) were used to deposit the coatings via constant voltage EPD. The distance between the electrodes in the EPD cell in all tests was kept constant at 10 mm. Deposition voltages and times were chosen in the range of 15–40 V and 15–120 min, respectively. Before deposition, substrates were cleaned and dried for 24 h at room temperature. Deposition voltages and times were studied in the ranges of 30–40 V and 30–120 min, respectively.

### 2.4. Characterization Methods

To determine the crystallographic structure of powders and deposited layers, X-ray diffraction (XPERT-PRO Diffractometer, Philips, Eindhoven, The Netherlands) with the wavelength 0.15406 nm was used. A scanning electron microscope (SEM, Philips 30 Xl, Eindhoven, The Netherlands) and energy-dispersive X-ray spectroscopy (EDX) were employed in order to evaluate the morphology and chemistry of the samples. A contact angle test was conducted to investigate the surface hydrophobicity. A surface roughness tester was used to determine the surface topology. A stereo microscope was also used to investigate the topology of the deposited layers. An MTT (3-(4,5-dimethylthiazol-2-yl)-2,5-diphenyltetrazolium bromide) test was employed to assess the biocompatibility and cell viability of synthesized samples. This test measures the shift in the MTT color of alive cells' mitochondria from yellow to purple formazan crystals, providing a photometric estimate of viable cell numbers. This test was conducted based on the ISO 10993-5:2009 standard [27]. The equipment utilized for the MTT test includes a laminar hood, a $CO_2$ incubator, and a centrifuge. Prior to MTT evaluations, samples were sterilized in the autoclave. The control sample consists of untreated wells without nanoparticle solutions. To determine cytotoxicity, the MTT test involves seeding $1 \times 10^4$ MG63 cell samples with DMEM culture media (containing 10% fetal bovine serum, 1% penicillin, and streptomycin) into 96-well plates in triplicate. Different concentrations (100, 75, 50, 25, and 10 μm) of the nanoparticle solution were added to the wells three times and incubated in a 5% carbon dioxide atmosphere at 37 ± 1 degrees Celsius for 24 h. After treatment, each well received 20 microliters of 5 mg/mL MTT dye. Three hours later, the MTT-color was extracted, and DMSO-solvent was applied to dissolve the purple crystals formed. An ELISA reader quantified the dye dissolved in the DMSO solvent. Wells with living cells exhibited higher optical density (OD) as compared to those with dead cells.

## 3. Result and Discussion

### 3.1. XRD Results

Figure 1 depicts XRD patterns BG-ZnO samples with different amounts of ZnO, ranging from 5 to 20%. The BG-ZnO powder mixture contains an amorphous bioactive glass phase and a crystalline ZnO phase. The former can be ascribed to a hill-type pattern (extending from 20 to 30 degrees), while the latter can be justified with sharp peaks. ZnO-related peaks appear at 2θ angles of 31.77, 34.41, 36.26, 47.54, 56.61, 62.85, 66.39, 67.96, 69.10, 72.56, 76.97, 81.38, and 89.62 degrees, corresponding to crystallographic planes (0 1 0), (0 0 2), (0 1 1), (0 1 2), (1 1 0), (0 1 3), (0 20), (1 1 2), (0 2 1), (0 0 4), (0 2 2), (0 1 4), and (0 2 3), respectively. Mentioned XRD peaks perfectly match with the JCPDS card number 96-901-1663. The crystallite size of ZnO nanoparticles, calculated using Scherer's equation [28], was 40 nm. It is also to be expected that the higher the ZnO content of the composite samples is, the sharper are the ZnO-related peaks as compared with the background.

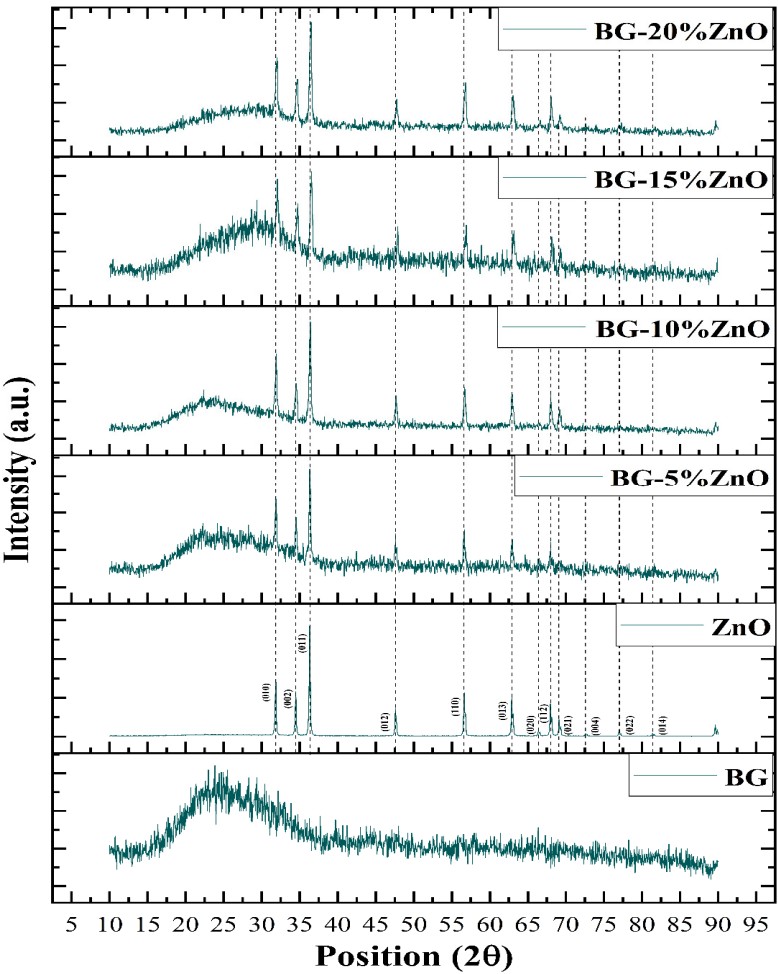

**Figure 1.** XRD patterns of bioactive glass (BG), ZnO particles, and ZnO-containing bioactive glass samples.

### 3.2. Stereomicroscope Results

Figure 2 depicts stereomicroscope images of electrophoretic bioactive glass coatings at different deposition times. The coatings were applied on the AISI 316L steel substrate with a constant voltage of 15 volts and a variable deposition time of up to 70 min. In this figure, the purple color is ascribed to the stainless-steel substrate, while the yellow color is related to the bioactive glass. As can be seen, initial nuclei of bioactive glass at the surface appear after 25 min of deposition. The increase in the deposition time from 25 to 40 min is accompanied by the formation of higher bioactive glass nuclei and the coalescence of existing bioactive glass islands. The higher the deposition time is, the more interconnected

the bioactive glass islands are, in such a way that after 50 min of deposition, more than 50% of the surface is covered with the deposited bioactive glass. A further increase in the deposition time up to 70 min ends up in a surface almost fully covered with a bioactive glass layer.

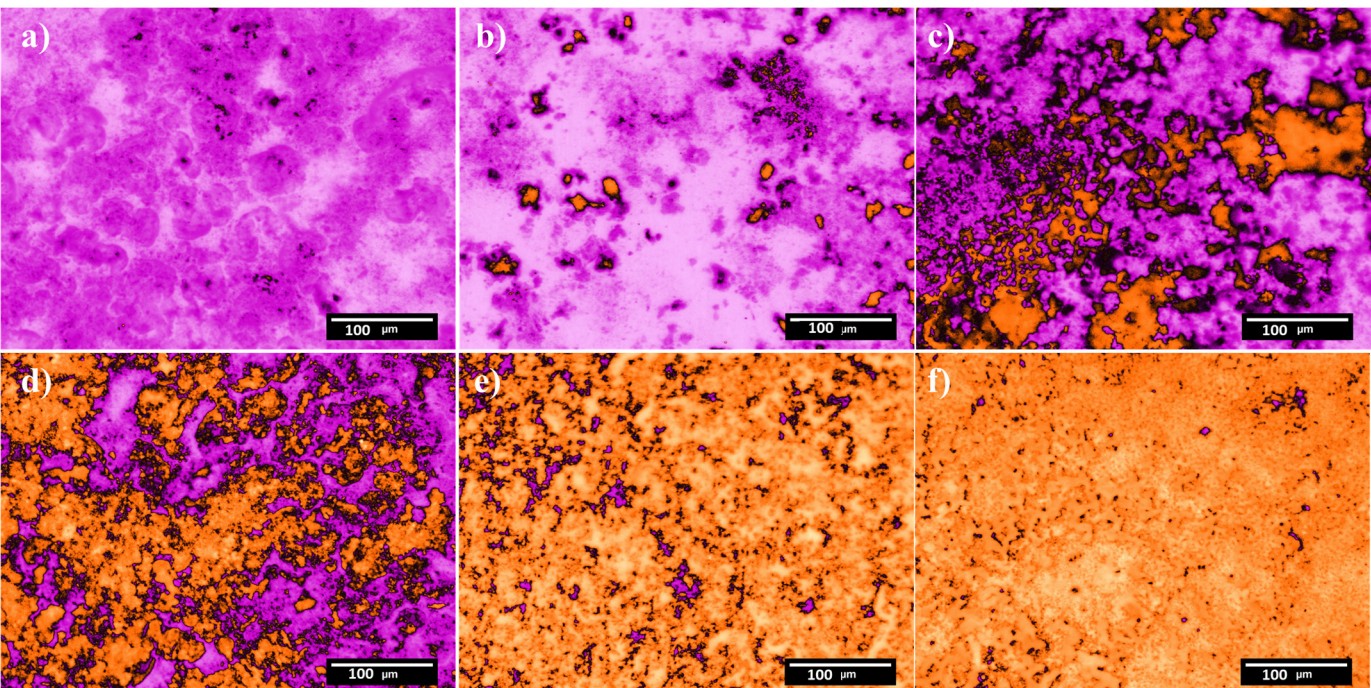

**Figure 2.** The stereomicroscope micrographs for surface deposited bioactive glass on AISI 316L substrate at constant voltage (15 V) at different times: (**a**) 0, (**b**) 25, (**c**) 40, (**d**) 50, (**e**) 60, and (**f**) 70 min. Note that the purple is the steel substrate and the yellow is the bioactive glass.

Figure 3 represents the stereomicroscope images of the 70 min-deposited bioactive glass/zinc oxide composites, with varying ZnO percentages from 5 to 20%. Overall, in all compositions 70 min appears to be sufficient to attain a uniformly deposited layer, with some slight differences in the morphology of the deposited layer. More precisely, the uncoated substrate in sample 15-ZB3 (15 wt.% ZnO and 85 wt.% BG) is in the form of continuous grooves (see Figure 3c), while in other samples, it is more in the form of scattered spots. Off course, one can expect that a higher deposition time (over 70 min) can result in a full coverage of the surface. Yet, the idea was to stop deposition before a complete surface coverage is attained to make sure that deposited layers have a low percentage of porosity. This porous surface structure is expected to have positive implications for the integration of the tissue into the deposited layer.

### 3.3. Contact Angle

Wettability is an extremely important surface characteristic [29]. A contact angle goniometer was utilized to measure the contact angle of deposited layers. As anticipated, an increase in the surface roughness is associated with a decrease in the wetting angle, indicating a higher degree of hydrophilicity. Figure 4a displays that the wetting angle of the SBF droplet on the AISI 316L substrate is 72°. The bioactive glass deposition overall ends up in a significant reduction in the wetting angle. For instance, the surface with a 25 min deposition has a wetting angle of 36° (see Figure 4b). A further increase in the deposition time up to 30 and 40 min results in a smaller angle down to approximately 28° (as depicted in Figure 4c,d). For deposition times higher than 40 min, the angles were too small to be measured, suggesting that the 70-min coated samples are superhydrophilic.

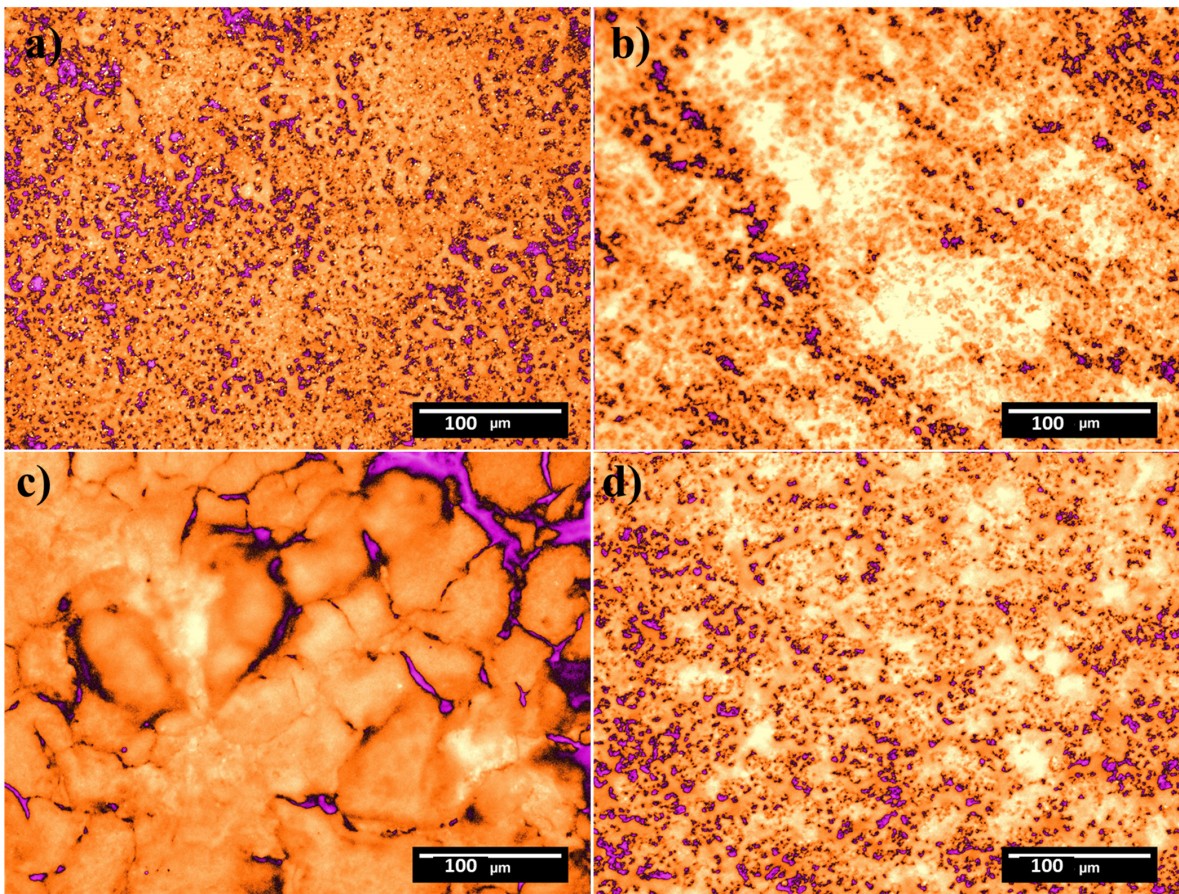

**Figure 3.** The stereomicroscope images for BG-ZnO composite coatings after 70 min of deposition in different concentrations of ZnO: (**a**) 5, (**b**) 10, (**c**) 15, and (**d**) 20 wt.%. Note that the purple is the steel substrate and the yellow is the bioactive glass.

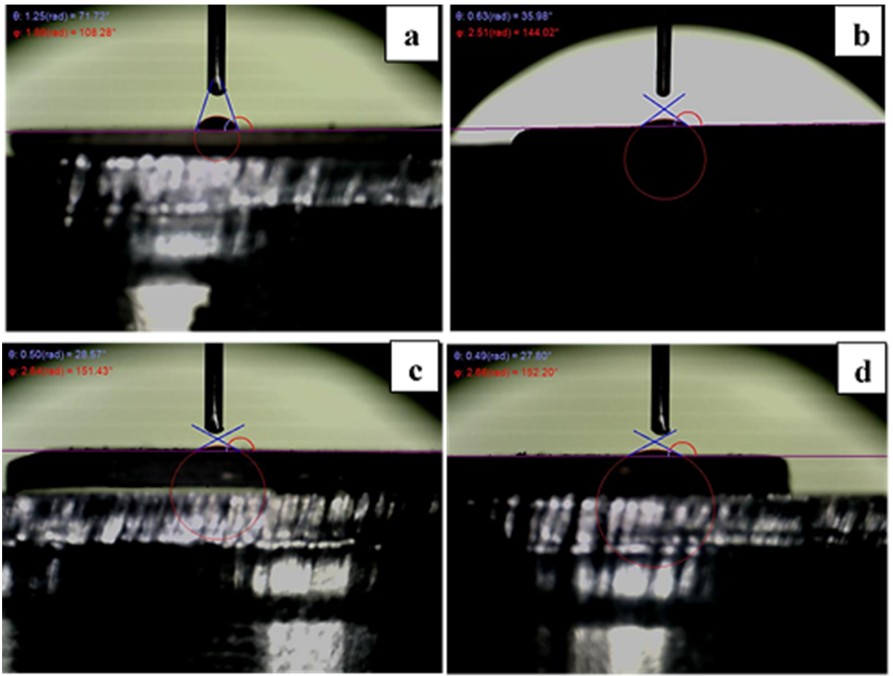

**Figure 4.** The results of contact angle tests for samples (**a**) the substrate without coating, (**b**) BG coating (25 min), (**c**) BG coating (30 min), and (**d**) BG coating (40 min).

### 3.4. Surface Roughness Measurements

Figures 5–7 depict the results of the roughness measurements, conducted on the coated bioactive glass coating at various time intervals from 25 to 70 min and BG-ZnO samples. In these figures, the red color is an indication of areas of higher roughness. Results show that the surface roughness increases up to 50 min of deposition, followed by a relatively slight decrease in the roughness up to 70 min (see Figures 5 and 7a). Moreover, it is apparent that the difference between the lowest valley and the highest peak increases from samples with a deposition time of 25 to 50 min, followed by a decrease thereafter. It is known that the higher the surface roughness is, the lower is the hydrophobicity of the surface. This argument is in agreement with the results of wettability tests in which surfaces with a lower roughness (with deposition times of 25, 30, and 40 min) exhibit some wettability angles between the droplet and the surface. Figures 6 and 7b illustrate the results of a similar test conducted on BG-ZnO composite coatings with varying percentages of ZnO, ranging from 5 to 20%. This test was performed under the same conditions as the previous one. The outcomes of this test reveal that ZnO-containing surfaces are comparatively rougher, as compared with bioactive glass coatings. Additionally, it is seen that the average roughness of these coatings increases with the increasing ZnO content up to 15%, followed by a dramatic decrease in 20%-ZnO composite. It is noteworthy that in all ZnO-containing samples, there is no measurable contact angle and all ZnO-containing surfaces can be considered as superhydrophilic.

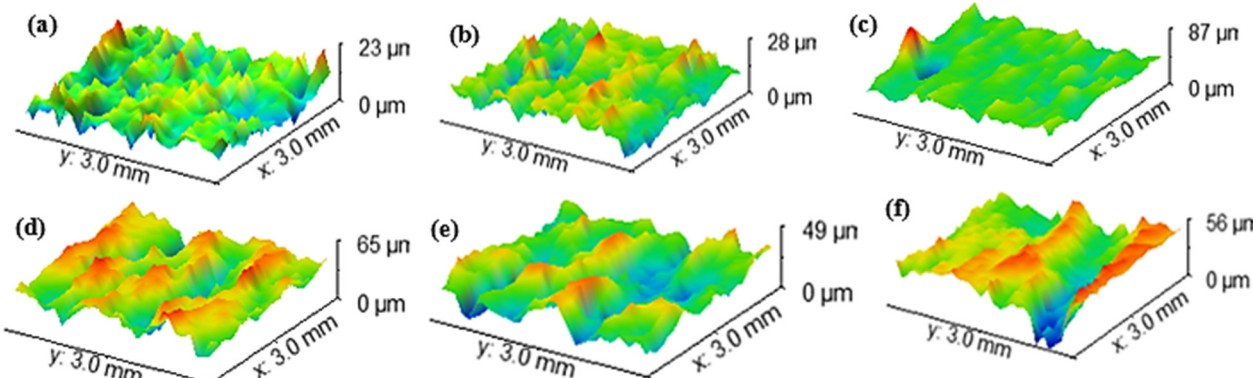

**Figure 5.** Laser profilometry of deposited bioactive glass coatings with different deposition times: (**a**) 25, (**b**) 30, (**c**) 40, (**d**) 50, (**e**) 60, and (**f**) 70 min.

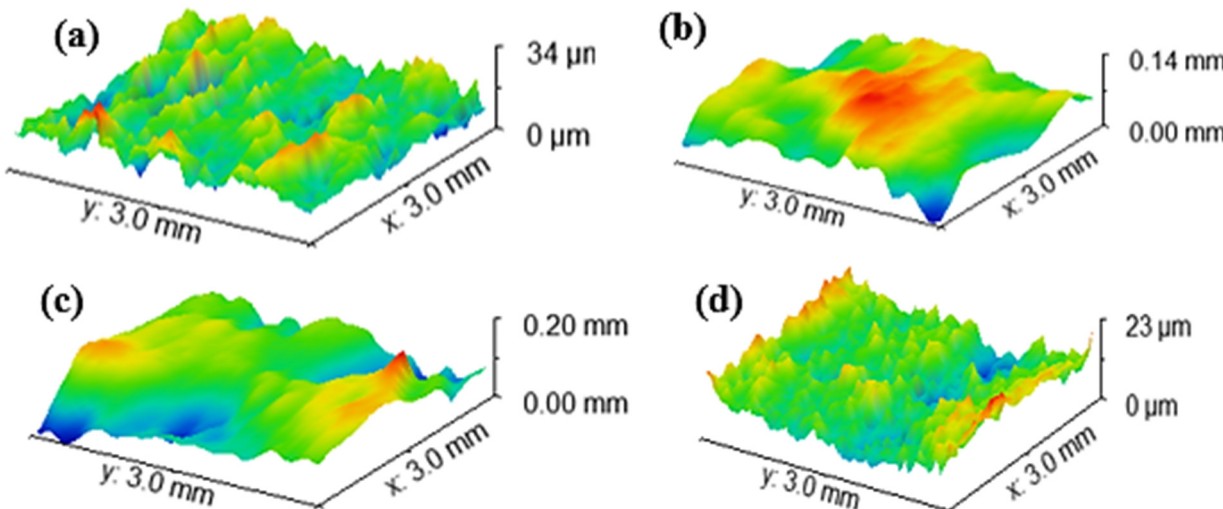

**Figure 6.** Laser profilometry of BG-ZnO composite coatings: (**a**) BG-5%ZnO, (**b**) BG-10%ZnO, (**c**) BG-15%ZnO, and (**d**) BG-20%ZnO.

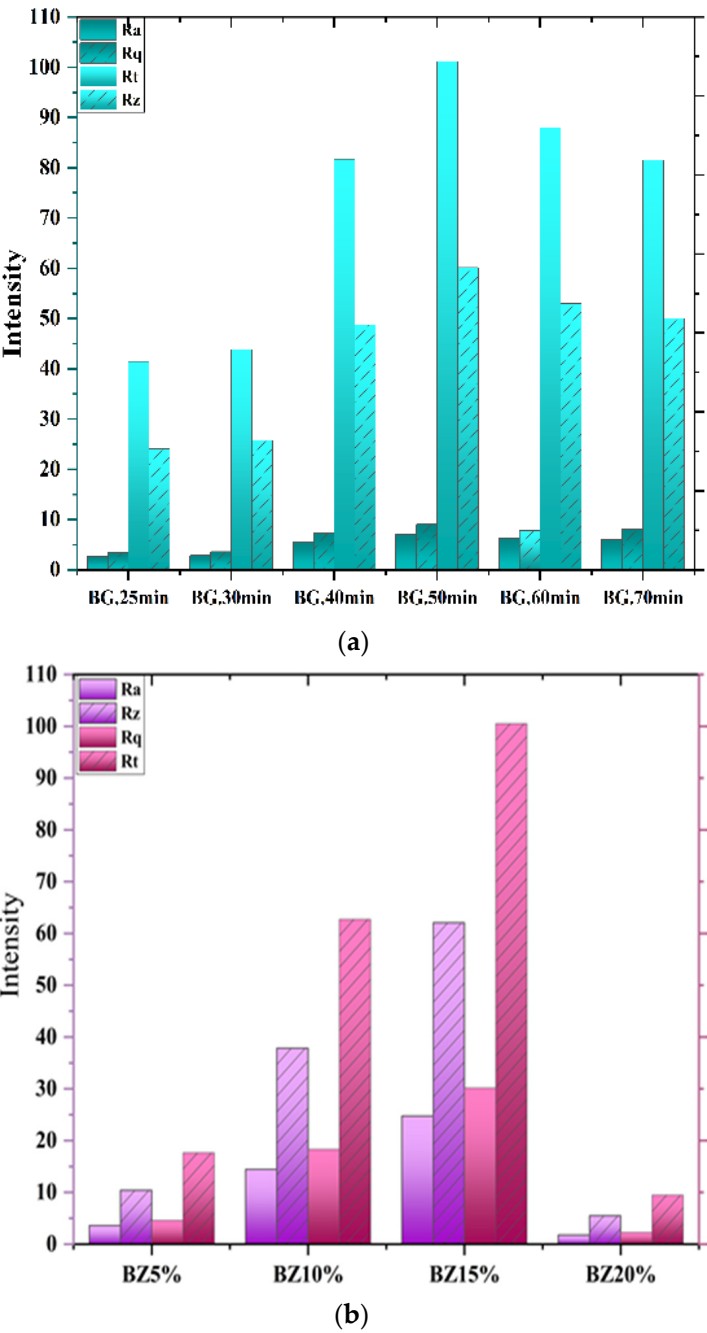

**Figure 7.** Surface roughness result of (**a**) bioactive glass for 25 to 70 min and (**b**) BG-ZnO coatings for 70 min (the standard deviation for the measurements is 5%).

*3.5. SEM/FESEM/EDS Analyses*

SEM micrographs of the deposited bioactive glass layer, as presented in Figure 8, demonstrate that increasing the deposition time up to 70 min leads to a more uniform coating. As can be seen in Figure 8a, 25 min of deposition results in a deposition of islands of bioactive glass (an example is pointed out by an arrow), indicating the non-uniformity of the coating layer. A further increase in the deposition time up to 40 min is associated with an improvement in the uniformity of the deposited layer. Ultimately, as depicted in Figure 8c, with a further increase in the deposition time, a more uniform layer with hardly any cracking in the coating is attained.

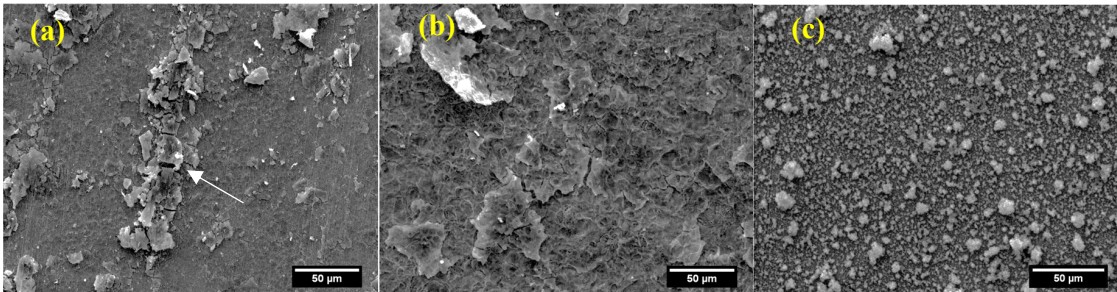

**Figure 8.** SEM micrographs of deposited bioactive glass in three different times: (**a**) 25, (**b**) 40, and (**c**) 70 min.

Figure 9 depicts SEM micrographs, EDS analysis and elemental mapping results of ZnO-containing bioactive glass layers, with the deposition times of all samples being 70 min. EDS results confirm the presence of elements such as silicon and calcium, characteristic of bioactive glass 58S5, in the coating as well as the uniform distribution of Zn inside the coating. Already mentioned arguments are well supported by the elemental mapping results. Clearly, as the percentage of Zn in the composition increases, the corresponding peaks in the EDS spectrum will become sharper, indicating a larger content of Zn elements in the coating. More importantly, EDS/elemental mapping results confirm a perfectly uniform distribution of ZnO in all samples, and that in turn indicates a successful incorporation of zinc inside the composite layer [29].

### 3.6. Corrosion Tests

Corrosion tests were conducted on samples using a conventional three-electrode system. The reference electrode used was Ag/AgCl, the counter electrode was a platinum auxiliary electrode, and the working electrode was made of the substrate. The electrochemical behavior of the samples in a simulated body fluid (SBF) solution, which was open to the air, was evaluated using electrochemical impedance spectroscopy (EIS) after a 24 h immersion period at $37 \pm 1$ °C. The frequency response analyzer was utilized to measure the frequency range from 1 (Hz) to 50 (kHz). The coating had an exposed surface area of 100 (mm$^2$) and the sinusoidal voltage signal had an amplitude of 10 (mV). After the EIS measurement, a potentiodynamic polarization test was conducted. The scan rate was 1 (mV.s$^{-1}$) and the scan range was from $-250$ (mV) concerning the open circuit potential (OCP) to $+1$ (V) versus Ag/AgCl.

### 3.7. Electrochemical Impedance Spectroscopy (EIS) Results

The Nyquist and Bode phase plots depicting the experimental and fitted results for both uncoated and coated samples are presented in Figure 10a,b, respectively. Upon analyzing the Nyquist plots, it is evident that the semicircle diameter of the uncoated base sample is greater than that of all the other coated samples. Particularly, the coated sample containing 20 wt.% zinc oxide particles exhibits the smallest semicircle, indicating its weakest barrier performance. Moreover, an increase in the coating corrosion and ion release is observed with an increase in the concentration of zinc oxide nanoparticles. It is widely recognized that a smaller diameter of the capacitive loop in the Nyquist plots corresponds to a lower polarization resistance, thereby supporting the previously obtained results [30–32]. Similar trends are observed in the Bode phase plots (Figure 10b). Notably, for the sample containing 20 wt.% zinc oxide nanoparticles in the coating, the phase angle at a frequency of 1 (Hz) is approximately 10 degrees, indicating severe corrosion and significant dissolution of the coating components in the solution. As the weight percentage of zinc oxide nanoparticles in the coating decreases to 15, 10, and 5, both the corrosion resistance and the phase angle increase. Among the tested samples, the AISI 316L stainless steel sample showed the largest phase angle in this series of graphs. At a frequency of 1 (Hz), the phase angle exceeds 75 degrees, indicating superior corrosion behavior, enhanced resistance to charge transfer, and improved corrosion performance compared to the other coated samples.

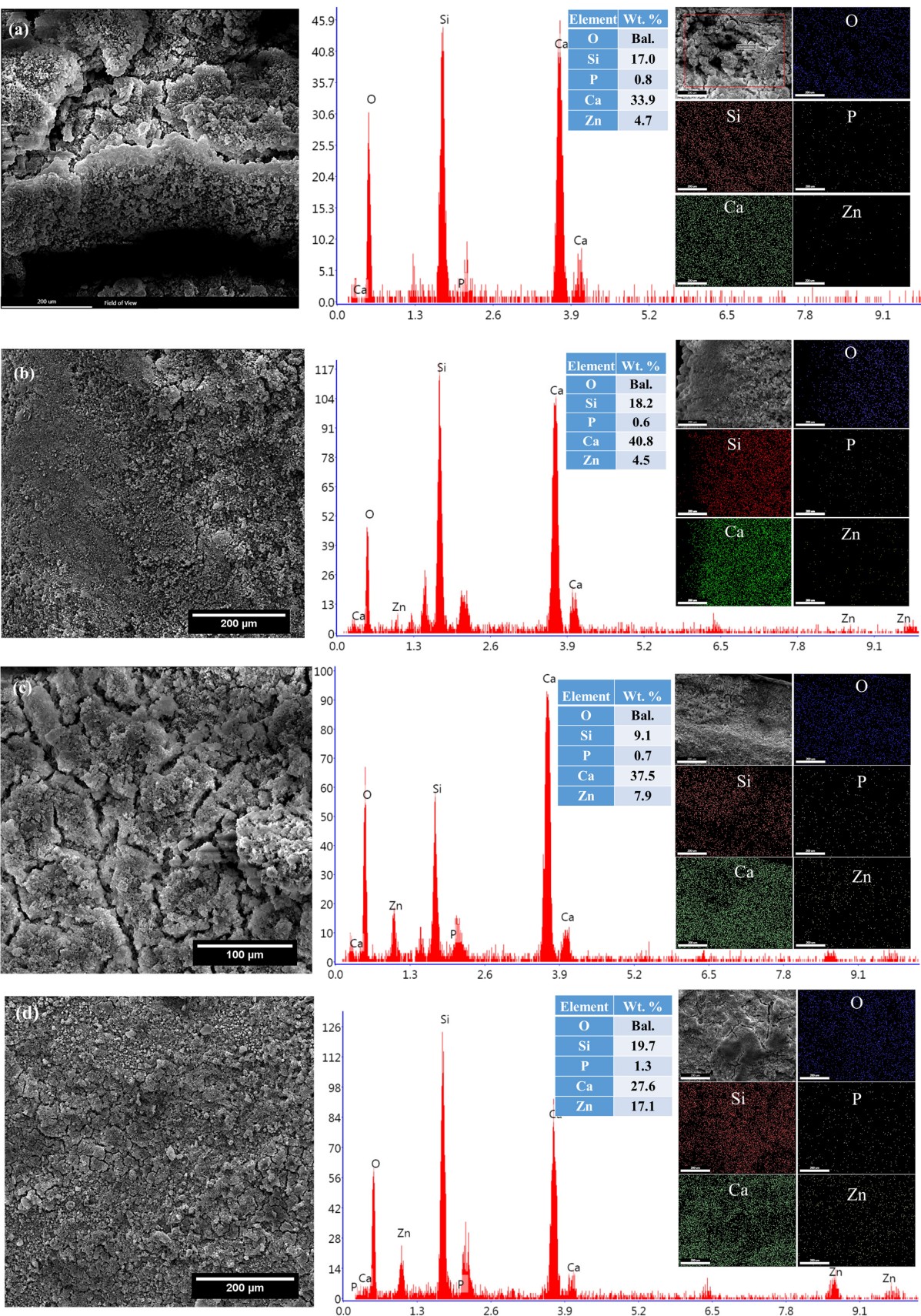

**Figure 9.** Result of SEM/EDS/elemental mapping analyses for (**a**) BG-5%ZnO, (**b**) BG-10%ZnO, (**c**) BG-15%ZnO, and (**d**) BG@-20%ZnO samples.

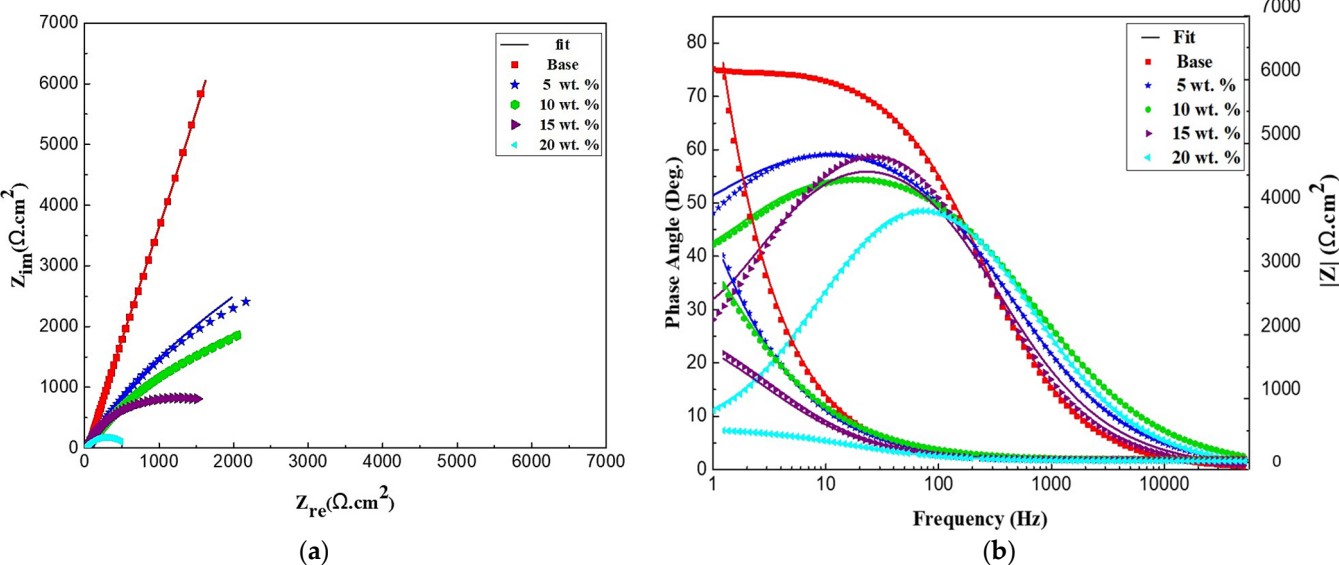

**Figure 10.** (**a**) Nyquist plots of experimental data (indicated by symbols) and their corresponding fitted results (indicated by solid lines across the symbols) after the samples were immersed in SBF solution for 24 h at $37 \pm 1$ °C. (**b**) Bode phase and Bode Z plots of experimental data (indicated by symbols) and their corresponding fitted results (indicated by solid lines across the symbols) after the samples were immersed in SBF solution for 24 h at $37 \pm 1$ °C.

An equivalent circuit (EC) model (Figure 11) based on a two-layer structure dielectric (an inner compact layer and an outer relatively porous layer) was used for fitting the experimental data of the substrates [33] and the coatings [34,35] using ZVIEW software (2006). This EC consisted of (CPEin-Rin) elevated by the barrier role of the oxide film primarily formed on a stainless steel substrate, and (CPEout-Rout) resulting from the porous outer layer of the passive film, which is decreased considerably by the presence of zinc oxide nanoparticles in the coating, and Rs related to solution resistance; the support for using an EC with two (CPE-R) combinations comes from the careful examination of the Bode phase plots. Overall, the corrosion resistance of stainless steels and coating materials is due to an inter-relation of several factors, including the chemistry of the alloy, the surface roughness, and the microstructure [36–38].

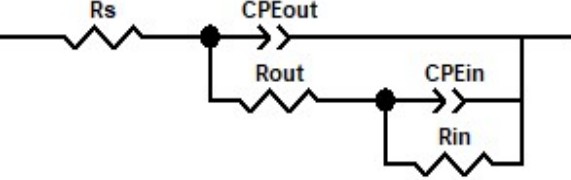

**Figure 11.** Equivalent circuit model applied for fitting impedance.

Table 1 confirms the good fitting results, with minor data errors (Chi-squared) supporting the findings. The presence of zinc oxide particles in the coating correlates with a decreased solution resistance. For AISI 316L stainless steel, the solution resistance measures 57.28 ($\Omega$ cm$^2$), while the sample containing 20 wt.% zinc oxide particles records 22.66 ($\Omega$ cm$^2$). This decrease in resistance can be attributed to the release of zinc oxide ions from the coating into the solution. A reduction in the content of zinc oxide in the coating leads to a significant increase in resistance. The AISI 316 stainless steel sample shows the highest internal and external resistances at 4213 (k$\Omega$ cm$^2$) and 89,118 ($\Omega$ cm$^2$), respectively. Conversely, the values of the constant phase element (CPE) follow the opposite trend, with the sample containing 20 wt.% zinc oxide displaying the highest CPE value. This suggests an increase in charge transfer and a corresponding rise in corrosion behavior.

**Table 1.** Electrical elements were extracted by fitting EIS data using EC in Figure 11. (Rs: resistance of solution, CPE: constant phase element, Rin and Rout: resistance of inner and outer layer, Chi-Squared: a measure of the goodness of fit between observed values and theoretically expected values, n: indicating the deviation from the ideal capacitor; $+1 \geq n \geq -1$).

| Sample | $CPE_{out}$ ($\mu F/cm^2$) | $R_{out}$ ($\Omega\ cm^2$) | n (Deviation from the Ideal Capacitor) | $CPE_{in}$ ($\mu F/cm^2$) | $R_{in}$ ($k\Omega\ cm^2$) | n | $R_s$ ($\Omega\ cm^2$) | Chi-Squared |
|---|---|---|---|---|---|---|---|---|
| Base | 31.78 (0.5) | 89,118 (940) | 0.86 (0.01) | 1.95 (0.15) | 4213 (462) | 0.98 (0.01) | 57.28 (3.7) | 0.001 |
| 5 wt.% | 67.01 (1.8) | 9752 (332) | 0.73 (0.01) | 101.5 (5.5) | 16.73 (1.8) | 0.79 (0.02) | 46.65 (1.8) | 0.0003 |
| 10 wt.% | 69.73 (5.15) | 6620 (181) | 0.69 (0.01) | 149.18 (3.8) | 13.21 (1.5) | 0.98 (0.02) | 42.83 (2.1) | 0.0001 |
| 15 wt.% | 72.78 (1.7) | 2247 (76.2) | 0.76 (0.01) | 550.07 (20.1) | 3.01 (0.29) | 0.91 (0.01) | 39.57 (1.9) | 0.002 |
| 20 wt.% | 88.84 (1.6) | 509.1 (14.7) | 0.76 (0.02) | 10,487 (315) | 0.91 (0.05) | 0.66 (0.01) | 22.66 (1.5) | 0.00002 |

*3.8. Potentiodynamic Polarization Test*

Following EIS tests, a potentiodynamic polarization test was conducted to further evaluate the sample behavior, and the results are presented in Figure 12, with the data presented in Table 2. The findings from this test confirm the findings from the EIS test. The AISI 316L stainless steel sample exhibits the lowest corrosion rate (5.91 $\mu A/cm^2$) and the least corrosion tendency (corrosion potential of $-0.34$ V). However, upon coating the sample and increasing the percentage of zinc oxide particles, both the corrosion rate and the tendency notably increase. For the AISI 316L stainless steel samples coated with 5 and 10 wt.% zinc oxide particles, an active–passive behavior is observed. At an increased potential of approximately 0.1 V, a passive layer forms, resulting in a decreased current density. However, at higher potentials of approximately 0.8 V, the current density rises again, indicating the loss of the protective layer. In the samples containing 5 and 10 wt.% zinc oxide particles, this behavior occurs at lower potentials and higher current densities. Conversely, the Tafel test results show no evidence of the formation of a passive layer in the samples containing 15 and 20 wt.% zinc oxide nanoparticles in the coating.

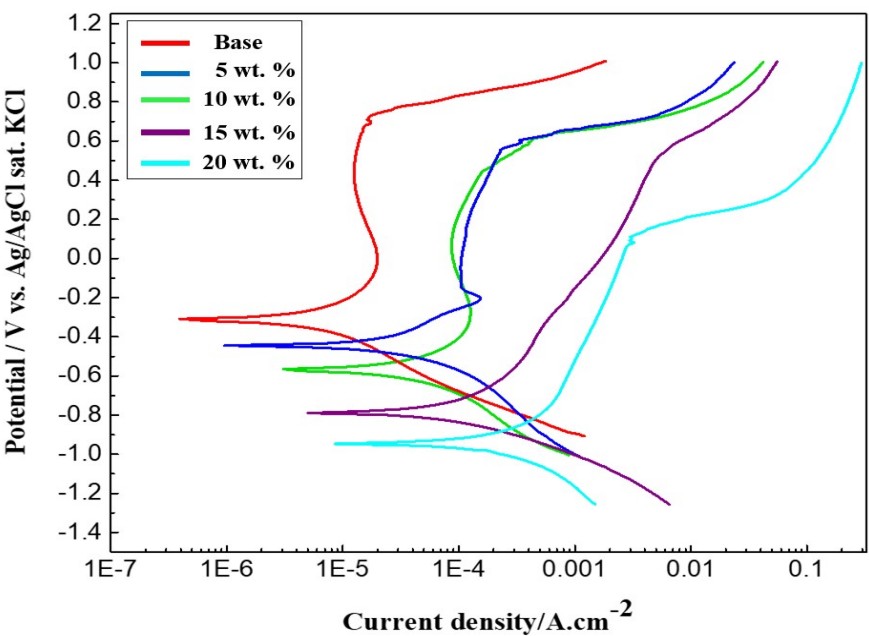

**Figure 12.** Potentiodynamic polarization curves in SBF solution at $37 \pm 1$ °C.

**Table 2.** Corrosion current density ($I_{corr}$) and corrosion potential ($E_{corr}$) for AISI 316 stainless steel, and coatings containing 5, 10, 15, and 20 wt.% zinc oxide nanoparticles in SBF solution at 37 ± 1 °C.

| Sample Name | E (V) | I (μA.cm$^2$) |
|---|---|---|
| Base | −0.34 (0.03) | 5.91 (0.4) |
| 5 wt.% | −0.39 (0.06) | 14.62 (0.9) |
| 10 wt.% | −0.62 (0.04) | 21.51 (1.4) |
| 15 wt.% | −0.78 (0.05) | 129.15 (7.2) |
| 20 wt.% | −0.92 (0.08) | 278.25 (12.4) |

*3.9. MTT Test Results*

Admission criteria for the MTT tests were defined as follows (Parts A to D):

**A: Viability Threshold for Cytotoxicity Determination:** A test sample is non-cytotoxic if its viability exceeds 70%; otherwise, it is classified as cytotoxic.

**B. Viability Assessment for Cellular Activity:** The examination of sample viability provides insights into cell growth and division in the vicinity of each test sample.

**C. Requirement for Control Sample Viability:** The control sample must exhibit 100% viability, indicating optimal conditions for cell survival and growth.

**D. Criterion for Standard Deviation:** To ensure robustness and consistency, the standard deviation between repeated samples within each experimental group should be below 15%, serving as a measure of reproducibility and precision.

The survival percentage of cells in contact with the analyzed sample was compared to the control sample, and the results show cytotoxicity levels of steel and bioactive glass samples by assessing cell survival percentages around them. The survival percentages of cells near the steel sample were measured in five different concentrations with three repetitions for a 2 h duration, revealing no cytotoxicity at any concentration level. Similarly, the survival percentages of cells near the bioactive sample were evaluated in five different concentrations with three repetitions after 2 h, consistently showing no cytotoxicity in all concentrations. Furthermore, the bioactive composite sample containing 10% zinc oxide was examined, and after 2 h, no cytotoxicity was observed in any of the five different concentrations with three repetitions. Additionally, the bioglass composite sample containing 20% zinc oxide showed no cytotoxicity in all concentrations after the 2-hour period, as assessed under five different concentrations with three repetitions [39]. Figure 13 also shows the cell viability in the control sample, AISI 316L, bioactive glass and BG-ZnO composites, containing 10% and 20% zinc oxide in 50 micromolar gel. According to the results shown in Figure 13, the cell viability rate of the 20% ZnO-containing sample is significantly higher as compared with other samples. Overall, the lowest cell viability rate is attained in the bare sample.

*3.10. ICP Test Results*

Figure 14 shows the results of a calcium and phosphorus reduction in the SBF solution for BG-ZnO composite samples after 4 weeks of immersion in the solution. As can be seen in this figure, the amount of calcium and phosphorus in the SBF solution is 46 and 48 mg/dL, respectively, at the beginning. In all cases, there is a significant decrease in the calcium and phosphorus concentrations, as compared with the initial solution. This depreciation in Ca and P is an indication that the Ca and P have deposited from the solution on the surface [40]. The fact that the calcium content in the SBF solution in ZnO-containing samples is slightly higher than that in the pure BG sample is an indication of a greater tendency to corrosion in ZnO-containing specimens, which is in agreement with the results of corrosion experiments.

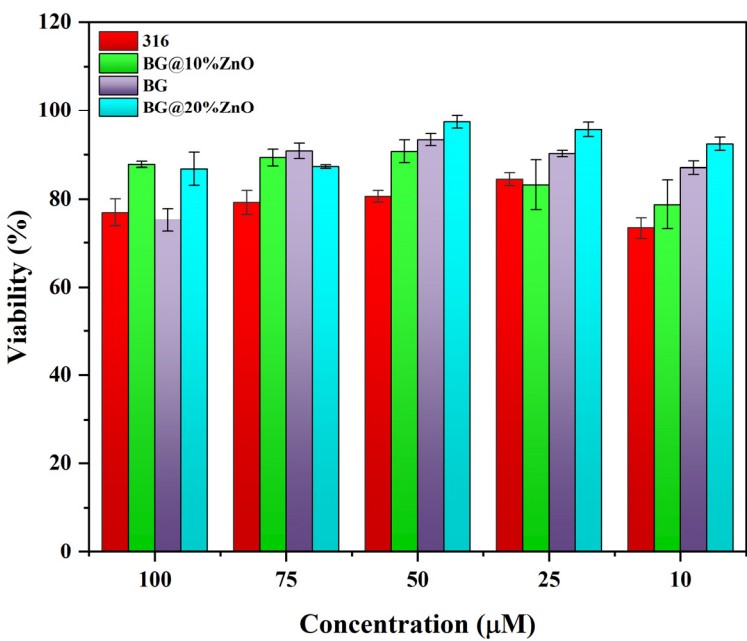

**Figure 13.** Viability of various samples in different concentrations.

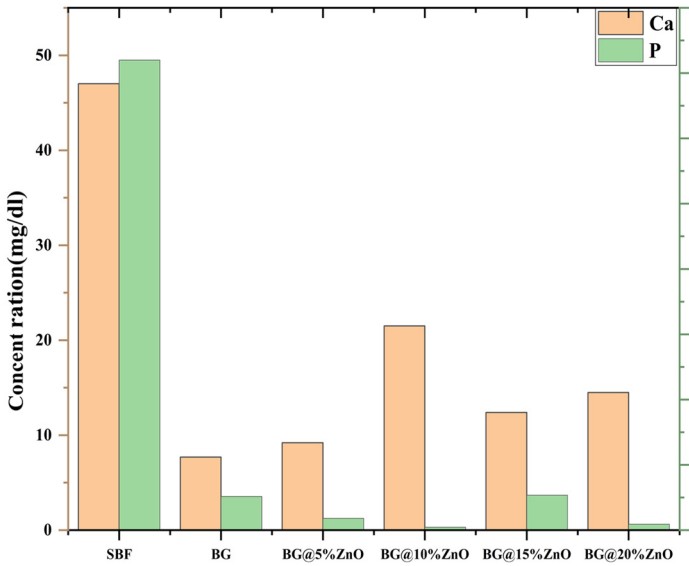

**Figure 14.** The ICP test result for Ca and P reduction in the SBF solution during 4 weeks of immersion (the measurement error is less than 5%).

### 3.11. Hydroxyapatite Crystals

The bioactivity of different compositions was investigated by evaluating the formation of hydroxyapatite crystals over the surface of samples. Results are displayed in Figure 15. Samples were immersed in 40 mL of SBF solution in a Bain-Marie at 37 degrees for 28 days. With the decrease in the content of calcium and phosphorus in the SBF solution (shown in Figure 14), it is anticipated that the hydroxyapatite crystals precipitate on the surface of samples. The formation of surface crystals can very well be confirmed by SEM/EDS investigations. Results show that the smallest amount of hydroxyapatite crystals forms over the surface of BG samples, whereas for ZnO-containing samples, hydroxyapatite covers nearly the entire surface. It appears that the higher the ZnO content is, the finer are the deposited hydroxyapatite crystals, which could be attributed to the difference in the roughness of the coating layers.

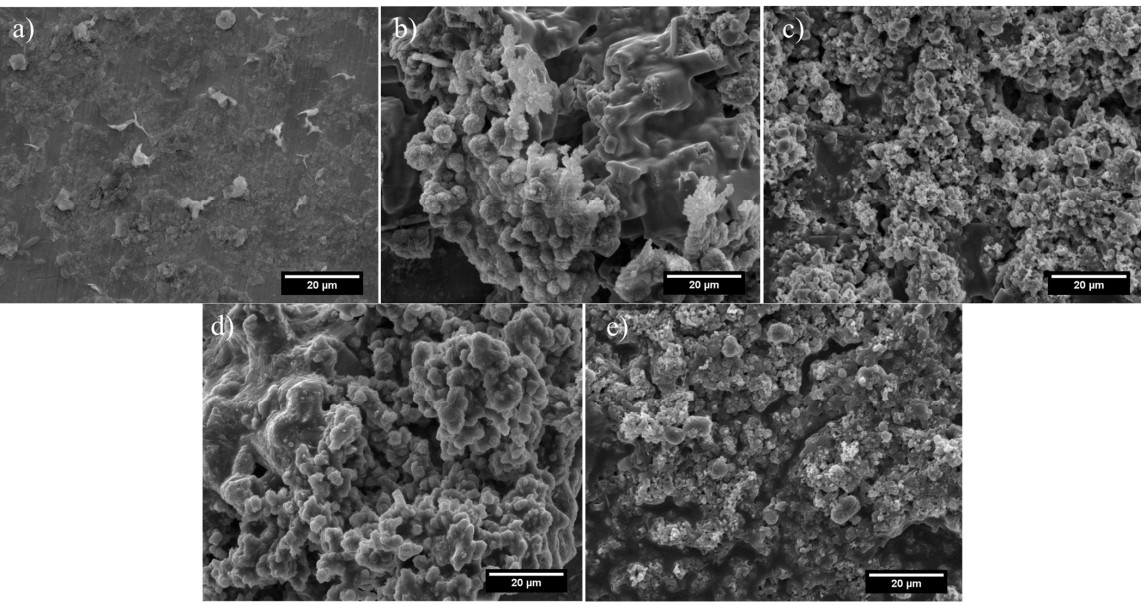

**Figure 15.** SEM results of bioactivity and hydroxyapatite formation for (**a**) bioactive glass sample, (**b**) BG@5%ZnO, (**c**) BG@10%ZnO, (**d**) BG@15%ZnO, and (**e**) BG@20%ZnO samples.

To further investigate the chemistry of crystals formed during immersion in SBF, EDS analysis was performed. Two typical analyses are given in Figure 16. Given that the chemical formula of hydroxyapatite is $Ca_{10}(PO_4)_6(OH)_2$ with a Ca/P ratio of 1.67, the formation of these crystals is very well confirmed.

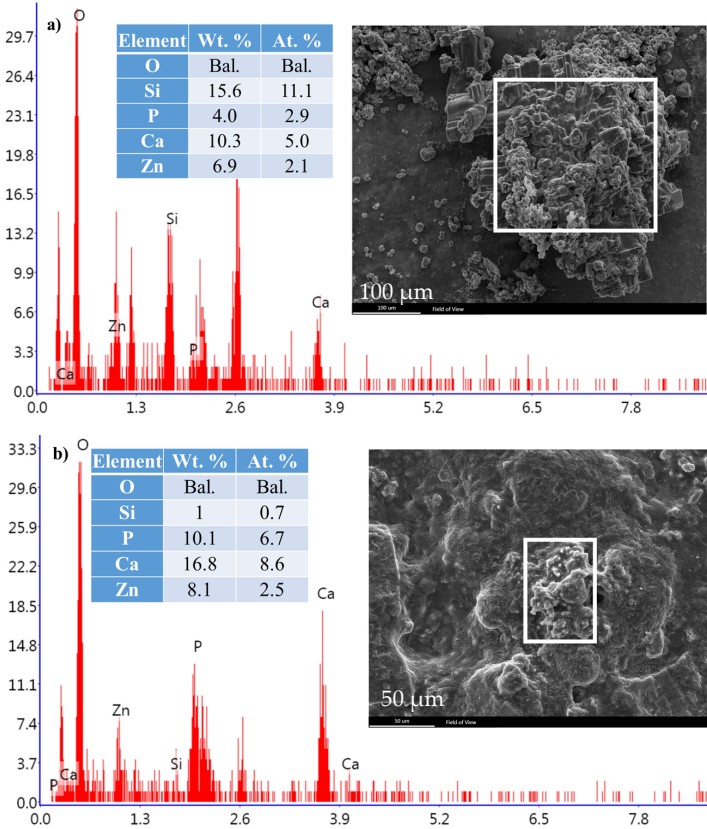

**Figure 16.** EDS analyses of crystals, formed after 4 weeks of immersion in SBF in samples (**a**) BG-5%ZnO and (**b**) BG-10%ZnO.

## 4. Conclusions

This study focuses on the synthesis of a bioactive glass-ZnO composite coating on the surface of AISI 316L stainless steel using an electrophoretic deposition (EPD) process. Comprehensive characterizations of deposited samples were performed, employing various analytical techniques. The surface functionalization of AISI 316L stainless steel is an extremely important issue when it comes to the biomedical application of this steel grade, given that stainless steels are not bioactive. The results of the characterization revealed a successful deposition of the zinc bioactive glass-ZnO layer using the EPD method. EDS analysis and elemental mapping showed a uniform distribution of elements (including zinc) over the surface. Results showed that the bioactive glass deposition overall results in a significant reduction in the wetting angle. For deposition times higher than 40 min, the wetting angles were too small to be measured, suggesting that deposited layers are superhydrophilic. In addition, it appears that ZnO-containing surfaces are comparatively rougher as compared with the bioactive glass layer. Results also showed that the higher the ZnO content is, the higher is the corrosion rate, suggesting that the presence of Zn in the coating increases the degradation rate of the coating layer. The cell viability and bioactivity of deposited layers increase with increasing ZnO contents, with the latter being confirmed by the formation of hydroxyapatite crystals on the surface after immersion in the SBF solution. Results showed that the bioactivity of the surface significantly increases with the incorporation of ZnO in the coating.

**Author Contributions:** Formal analysis, B.H.B.; Investigation, M.R., F.H.L. and M.R.K.; Data curation, M.S.A.; Writing—review & editing, M.R. and S.H.M.A.; Supervision, A.B. All authors have read and agreed to the published version of the manuscript.

**Funding:** This research received no external funding.

**Institutional Review Board Statement:** Not applicable.

**Informed Consent Statement:** Not applicable.

**Data Availability Statement:** Data are contained within the article.

**Conflicts of Interest:** The authors declare no conflict of interest.

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
