# Peer review of "Electrophoretic Deposition of ZnO-Containing Bioactive Glass Coatings on AISI 316L Stainless Steel for Biomedical Applications"

_coatings, doi:10.3390/coatings13111946_

Round 1

Reviewer 1 Report

Comments and Suggestions for Authors

The main objective of this study is to investigate the effect of incorporating zinc oxide nanoparticles into bioactive glass matrices on the biological activity and structural properties of deposited coatings. This paper is of great significance for the study of coatings. But there are some issues in the article that need to be revised:

1.   Mentioned in subsection 2.2:”First of all, for the …… at room temperature.” The authors are advised to add a table to make drug dispensing more intuitive.

2.   Mentioned in subsection 2.4:”To determine cytotoxicity …… plates in triplicate.” The abbreviation of the first occurrence is not explained, the author is asked to check and add it.

3.   Mentioned in subsection 3.2:”Fig.3 represents…..percentages from 0 to 20%.” The percentages represented in Figure 3 are not 0 to 20, please check and confirm by the authors.

4.   Mentioned in subsection 3.2:”15-ZB3 is in the form of …… see fig.3e.” The mixed solution of 15 percent zinc oxide and 85 percent BG corresponds to Figure 3c, which the authors were asked to check and confirm.

5.   Mentioned in subsection 3.3:”contact angle goniometer …… coated samples are……”The author describes the angle of change over time, but does not specify which solution to use, please the author to indicate.

6.   Mentioned in subsection 3.5:”As can be seen in …… of islands of bioactive glass,” Figure 8a shows the topography of 25 minutes of deposition, not 20 minutes, please check and confirm by the authors.

7.   Mentioned in subsection 3.10:”the amount of …… at the beginning.” The description does not match the data in Figure 14, please check and confirm by the authors.

Comments on the Quality of English Language

need improving

Author Response

Dear Reviewer,

Thanks so much for your comments and inputs. We found your comments very useful and informative and we did our best to revise the manuscript taking all your concerns into consideration. Here please find detailed explanation, related to our replies to each of your comments.

We hope you find it convincing.

Best Regards,

The main objective of this study is to investigate the effect of incorporating zinc oxide nanoparticles into bioactive glass matrices on the biological activity and structural properties of deposited coatings. This paper is of great significance for the study of coatings. But there are some issues in the article that need to be revised:

Thanks so much for the positive remark. Highly appreciated.

  1. Mentioned in subsection 2.2:”First of all, for the …… at room temperature.” The authors are advised to add a table to make drug dispensing more intuitive.

Thanks for your comment. The following part describes in detail the MTT test and composition of the solution used for the test. I suppose this will be enough.

To determine cytotoxicity, the MTT test involves seeding 1*104 MG63 cell samples with DMEM culture media (containing 10% fetal bovine serum, 1% penicillin, and streptomycin) into 96-well plates in triplicate. Different concentrations (100, 75, 50, 25, and 10 um) of the nanoparticle solution were added to the wells three times and incubated in a 5% carbon dioxide atmosphere at 37±1 degrees Celsius for 24 h. After treatment, each well receives 20 microliters of 5 mg/ml MTT dye.

  1. Mentioned in subsection 2.4:”To determine cytotoxicity …… plates in triplicate.” The abbreviation of the first occurrence is not explained, the author is asked to check and add it.

Thanks. This is added to the paper (Line 128).

  1. Mentioned in subsection 3.2:”Fig.3 represents…..percentages from 0 to 20%.” The percentages represented in Figure 3 are not 0 to 20, please check and confirm by the authors.

Sorry for the mistake. This is corrected in the manuscript (Line 182).

  1. Mentioned in subsection 3.2:”15-ZB3 is in the form of …… see fig.3e.” The mixed solution of 15 percent zinc oxide and 85 percent BG corresponds to Figure 3c, which the authors were asked to check and confirm.

True. This is corrected in the manuscript (Line 186).

  1. Mentioned in subsection 3.3:”contact angle goniometer …… coated samples are……”The author describes the angle of change over time, but does not specify which solution to use, please the author to indicate.

The solution is added to the paper (Line 200).

  1. Mentioned in subsection 3.5:”As can be seen in …… of islands of bioactive glass,” Figure 8a shows the topography of 25 minutes of deposition, not 20 minutes, please check and confirm by the authors.

Corrected, as per your note. Thanks so much.

  1. Mentioned in subsection 3.10:”the amount of …… at the beginning.” The description does not match the data in Figure 14, please check and confirm by the authors.

Corrected, as per your note. Thanks so much.

Reviewer 2 Report

Comments and Suggestions for Authors

The study investigated the effect of ZnO-containing active glass on AISI 316L stainless steel for implant applications. While the novelty of the research is commendable, there are concerns about the clarity of data representation. Here are some suggestions for improvement:

1.     BG- 20%ZnO (20 wt.% ZnO and 70 wt.% BG). Line 91-93. Please check again

2.     The microstructure in Fig. 2 was taken from which sample?

3.     Why are microstructures in BG-15ZnO different from the others? Please elaborate!

4.     Fig 3 shows the microstructure of samples after EPD for 70 min (stop experiment), Where is the microstructure after 120 min (as explained in the Experimental Method section)?

5.     Why do the authors choose to check the contact angle after EPD for 25, 30, and 45 minutes (Fig. 4)? While other data shows 70 min? Why the data from Zn-O containing was not provided? The data didn’t support the goal of this study.

6.     For Fig. 7b, how long is the duration of the EPD process? The information should be added to the figure caption

7.     Please improve the quality of the annotation in Fig. 10.

8.     For Bode plot in Fig. 11b, please provide impedance-frequency curve as well!

9.     What is the meaning of values in brackets found in Table 1 and 2? Authors need to clarify using foot notes (under the Tables) or in the Table captions.

10.  “This decrease in resistance can be attributed to the release of zinc oxide ions from the coating into the solution”. What is the consequence of decreasing corrosion resistance to the success of bio-implantation? Please elaborate!

11.  Why only provide EDS data for samples BG-5ZnO and BG-10ZnO? which composition is the best? express the final findings and insights in the "Conclusions" section.

Author Response

Dear Reviewer,

Thanks so much for your comments and inputs. We found your comments very useful and informative and we did our best to revise the manuscript taking all your concerns into consideration. Here please find detailed explanation, related to our replies to each of your comments.

We hope you find it convincing.

Best Regards,

The study investigated the effect of ZnO-containing active glass on AISI 316L stainless steel for implant applications. While the novelty of the research is commendable, there are concerns about the clarity of data representation. Here are some suggestions for improvement:

  1. “BG- 20%ZnO (20 wt.% ZnO and 70 wt.% BG).” Line 91-93. Please check again

Sorry for the mistake. This is corrected in the manuscript (Line 91-93).

  1. The microstructure in Fig. 2 was taken from which sample?

Thank you for your question. It was taken from taken from the coated bioglass surface in different time from 25 min to 70 min.

  1. Why are microstructures in BG-15ZnO different from the others? Please elaborate!

Thank you for your comment. Due to the circumstances in the EPD process. The thickness of its sample was increased and being different. That is why the surface structure is slightly different

  1. Fig 3 shows the microstructure of samples after EPD for 70 min (stop experiment), Where is the microstructure after 120 min (as explained in the Experimental Method section)?

Thank you for your question. Considering the fact that the structure after 70 minutes has had no impact on the uniformity of the coverage and has only led to an increase in the thickness of the coating, we only reported data till 70 min.

  1. Why do the authors choose to check the contact angle after EPD for 25, 30, and 45 minutes (Fig. 4)? While other data shows 70 min? Why the data from Zn-O containing was not provided? The data didn’t support the goal of this study.

Thank you for your question. Based on the profilometric graph, it can observed that the mentioned samples had smoother surface, allowing for the measurement of contact angle. However, with an increase in time and an increase in surface roughness, the hydrophilicity of the samples increases such a way that that is impossible to report the contact angle images.

  1. For Fig. 7b, how long is the duration of the EPD process? The information should be added to the figure caption.

Thank you for your comment, the time is added to the caption (Line 238 and 239)

  1. Please improve the quality of the annotation in Fig. 10.

Thank you for your comment. This is done in the manuscript (Line 298-305(.

  1. For Bode plot in Fig. 11b, please provide impedance-frequency curve as well!

Thank you for your comment. This is done in the manuscript.

  1. What is the meaning of values in brackets found in Table 1 and 2? Authors need to clarify using foot notes (under the Tables) or in the Table captions.

Thank you for your comment. This is done in the manuscript (Line 329-332 and 351)

  1. “This decrease in resistance can be attributed to the release of zinc oxide ions from the coating into the solution”. What is the consequence of decreasing corrosion resistance to the success of bio-implantation? Please elaborate!

Thank you for your comment. Considering that the application of these implants in osseous location, the formation of a hydroxyapatite structure and its soft formation are of utmost importance to us. Therefore, given that bioactive glass particles are used in our coating, the release of ions from this material in the environment accelerates the bone formation process and the formation of hydroxyapatite layer on the sample’s surface.

  1. Why only provide EDS data for samples BG-5ZnO and BG-10ZnO? which composition is the best? express the final findings and insights in the "Conclusions" section.

Thank you for your comment. These are given as typical results. As mentioned in the manuscript, the results of other samples are very similar to the reported data.

Reviewer 3 Report

Comments and Suggestions for Authors

Dear Author,

After a careful and thorough review, below are my considerations.

 Title is suitable for the research performed.

 HIGHLIGHT

The highlights are missing.

 ABSTRACT

The information presented here does not summarize the study that has been developed. It is necessary for the authors to provide more details about electrophoretic deposition, bioactive glass coatings, why AISI 316L stainless steel is used, which biomedical applications can be favored, and finally present the results of the study. The characterization techniques commented from line 16 to line 20 must be removed from the Abstract as this information must appear in the Materials and Methods section.

 1. INTRODUCTION

There is only 1 (one) paragraph here! I ask that the introduction have a few shorter paragraphs. Also, in lines 68 and 69, it says, "Different methods can be used to apply a bioactive glass-based coating layer to the surface of an implant". What are the other methods? An explanation is needed so that the reader knows why electrophoretic deposition (EPD) is more appropriate in this study.

2. MATERIALS AND METHODS

AISI 316L was used in the study as a matrix to receive the coating with ZnO-containing bioactive glass coatings. It is also necessary to provide information on the chemical composition of this material.

 3. RESULTS AND DISCUSSION

In Figure 1 (XRD patterns), the identification of the peaks needs to be improved, as in the printed version of the manuscript it is very difficult to observe the identification of these peaks.

Figure 7 does not show the standard deviation of each measurement.

In Figure 8, arrows must be added to make it easier for the reader to understand the authors' discussion.

In Figure 9, why is the EDS spectrum graph included? I ask the authors to remove the EDS graph and place the element quantification table within the respective SEM micrograph. This will allow more space to enlarge the mapping images, which are very small.

Figure 14 does not show the standard deviation of each measurement.

In Figure 15, insert arrows to facilitate the reader's understanding of the authors' discussion.

In Figure 16, follow the same procedure as in Figure 14. Enlarge the micrograph and also the scale, which is very small and impossible to see.

 5. CONCLUSIONS

It is not necessary to repeat in the conclusions what characterization techniques were used in the study. Here, based on the title and objectives presented, it is necessary for the authors to state the results of the study and the progress that the study represents in the field of health, in this case for implants. I ask that the authors take more time to gather information pertinent to the study and rewrite the conclusions.

 REFERENCES

The number of references is in accordance with the research conducted and the theoretical basis presented.

Author Response

REVIEWER 3

Dear Reviewer,

Thanks so much for your comments and inputs. We found your comments very useful and informative and we did our best to revise the manuscript taking all your concerns into consideration. Here please find detailed explanation, related to our replies to each of your comments.

We hope you find it convincing.

Best Regards,

ABSTRACT

The information presented here does not summarize the study that has been developed. It is necessary for the authors to provide more details about electrophoretic deposition, bioactive glass coatings, why AISI 316L stainless steel is used, which biomedical applications can be favored, and finally present the results of the study. The characterization techniques commented from line 16 to line 20 must be removed from the Abstract as this information must appear in the Materials and Methods section.

Reply: Many thanks for your comment. The abstract was completely revised as per your suggestion. Lines 16 to 20 were removed.

  1. INTRODUCTION

There is only 1 (one) paragraph here! I ask that the introduction have a few shorter paragraphs. Also, in lines 68 and 69, it says, "Different methods can be used to apply a bioactive glass-based coating layer to the surface of an implant". What are the other methods? An explanation is needed so that the reader knows why electrophoretic deposition (EPD) is more appropriate in this study.

Reply: Many thanks for your comment. The introduction was revised as per your recommendation. As per your comment, other deposition methods are enlisted and more information on the EPD was added to the manuscript.

  1. MATERIALS AND METHODS

AISI 316L was used in the study as a matrix to receive the coating with ZnO-containing bioactive glass coatings. It is also necessary to provide information on the chemical composition of this material.

Reply: Many thanks for your comment. The composition of the alloy was added to the manuscript.

  1. RESULTS AND DISCUSSION

In Figure 1 (XRD patterns), the identification of the peaks needs to be improved, as in the printed version of the manuscript it is very difficult to observe the identification of these peaks.

Reply: All angles are precisely mentioned in Line 154-157

Figure 7 does not show the standard deviation of each measurement.

Reply: The standard deviation for the measurements are 5%. This is added to the image title.  

In Figure 8, arrows must be added to make it easier for the reader to understand the authors' discussion.

Reply: Done, as per your comment.   

In Figure 9, why is the EDS spectrum graph included? I ask the authors to remove the EDS graph and place the element quantification table within the respective SEM micrograph. This will allow more space to enlarge the mapping images, which are very small.

Reply: Thanks for your comment. I do believe that EDS spectrum is an important data, as it provides a confirmation for the numbers, given in the table. Elemental mapping, is just a visual data. Putting all these data together is indeed a giving a package of data in one image, starting from SEM, and ending up in elemental mapping.

Figure 14 does not show the standard deviation of each measurement.

Reply: Thanks for your comment. The measurement error in this case is less than 5%.

In Figure 15, insert arrows to facilitate the reader's understanding of the authors'

Reply: Thanks for the comment. In this SEM micrograph series, all surface features are in fact HA crystals.

discussion.

In Figure 16, follow the same procedure as in Figure 14. Enlarge the micrograph and also the scale, which is very small and impossible to see.

Reply: Thanks for the comment. A larger scale bar was used in Figure 16.

  1. CONCLUSIONS

It is not necessary to repeat in the conclusions what characterization techniques were used in the study. Here, based on the title and objectives presented, it is necessary for the authors to state the results of the study and the progress that the study represents in the field of health, in this case for implants. I ask that the authors take more time to gather information pertinent to the study and rewrite the conclusions.

Reply: Thanks for the comment. The conclusion was re-written as per your recommendation.  

 REFERENCES

The number of references is in accordance with the research conducted and the theoretical basis presented.

Reply: Many thanks.

Round 2

Reviewer 3 Report

Comments and Suggestions for Authors

Dear Author,

I have reviewed the corrections made to the manuscript. The time spent on the corrections was important. The manuscript now meets the requirements to be considered for publication.
